# An Experiment in Denmark to Infect Wounded Sitka Spruce with the Rotstop Isolate of *Phlebiopsis gigantea*, and Its Implications for the Control of *Heterobasidion annosum* in Britain

**DOI:** 10.3390/pathogens11080937

**Published:** 2022-08-18

**Authors:** Jim Pratt, Iben M. Thomsen

**Affiliations:** 1Mountain Cross, West Linton EH46 7DF, Scotland, UK; 2Institut for Geovidenskab og Naturforvaltning, Københavns Universitet, Skov, natur og biomasse, Rolighedsvej 23, 1958 Frederiksberg C, Denmark

**Keywords:** *Heterobasidion annosum*, Sitka spruce, biocontrol, rotstop, *Phlebiopsis gigantea*, stem damage, Britain, field testing, quarantine

## Abstract

The formulation of a Finnish isolate of the saprotrophic wood-rotting fungus *Phlebiopsis gigantea* into the biocontrol agent (BCA) Rotstop, which is used to prevent infection of Norway spruce stumps by aerial basidiospores of *H. annosum,* has led to its application to more than 200,000 ha of forest in Scandinavia and North Europe. The success of this treatment opens the possibility of introducing the Rotstop strain into Britain for use on Sitka spruce stumps, which at present (2022) are lacking adequate prophylactic treatment. However, Rotstop is probably non-native to Britain and to North America (the ancestral home of this spruce), and we do not know if this xylem-decaying BCA can invade standing trees. In this paper, we describe a trial into this issue conducted for the U.K. Forestry Commission in Denmark, in a country where both Rotstop and Sitka spruce have been naturalised. It was preliminary to further stump treatment trials, and is relevant to long-term issues surrounding stump treatment in Britain. Inoculations into 44-year-old standing Sitka spruce with 20 mm wooden Scots pine plugs pre-colonised with Rotstop resulted in decay of the S1, S2, S3 and middle lamellae of sapwood above and below the wounds after 11–18 months. In contrast, infection of sapwood occurred in only one of 39 wounds on the same trees challenged with oidial spore inoculants adpressed to undamaged xylem sapwood during the same period. While the results suggest that release of Rotstop into the productive stands of Sitka spruce in Britain would be unlikely to cause long-term commercial losses to wounded trees, the work highlights issues relating to the assessment of risk associated with the introduction of non-native BCAs within the forest environment.

## 1. Introduction

Britain’s forests, decimated in two world wars, have been restructured in the past 100 years by replanting felled stands and planting new ones, mostly on land unsuitable for agriculture. No native species of conifer are suitable for this project, and the choice largely fell on Sitka spruce (*Picea sitchensis* (Bong). Carr) from coastal NW America. This species now accounts for more than 50% of all home-grown British timber and will continue to do so for the anticipated future following significant investment in its genetic improvement.

Sitka spruce is, however, very susceptible to *Heterobasidion annosum* (Fr.) Bref. s.l., and in the absence of a reliable, effective prophylactic stump treatment, can be expected to suffer losses through rejection of butt-rotted timber of up to 40% of value of each infected tree [1]. Significant research into the risk factors for this disease [2] allowed a process-model of its life cycle in stands of Sitka spruce to be used for predicting, on a site-to-site and a country-wide basis, the long-term prognosis of the disease in the U.K. over the next few hundred years [3]. What is clear is that the present low level of *H. annosum* within stands of Sitka spruce will only be maintained by the assiduous application of an effective prophylactic stump treatment in perpetuity. Of the three stump treatment materials available in Britain at the Millennium, only one (disodium octaborate tetrahydrate, or DOT) was effective on Sitka spruce stumps [4]. However, for administrative and regulatory reasons, DOT is no longer available, while neither of the two alternatives, urea and a British isolate of *Phlebiopsis gigantea* (Fr.) Jülich (namely, PG Suspension), provide consistent and reliable control on Sitka spruce.

It is relevant at this point to consider, briefly, an effect of the inundation of the North Sea and the creation of the English Channel soon after the last ice age (approximately 12,000 years ago). The channel effectively isolated Britain from recolonisation by many European species of plants, of which Norway spruce (*Picea abies* L.) was one, along with its suite of companion organisms which probably included the saprotrophic *P. gigantea* that has been used in Rotstop. Because of the youth of Britain’s forest estate, levels of disease incidence at present are low, especially compared to those in stands of Norway spruce in Scandinavia [5], where national inventories established incidence of decay caused by *Heterobasidion* spp. to vary between 9% and 30% of standing trees, destroying more than 2 million cubic metres of usable timber per year.

Given the clear long-term risks, finding a suitable long-term prophylactic stump treatment that is environmentally acceptable is clearly necessary. Rotstop [6], with its specific ability to grow and replace *H. annosum* spp. in stumps of Norway spruce, is an obvious candidate to test on Sitka spruce. However, although the species *P. gigantea* is native to Britain, the genotype of the BCA (biological control agent) used in Rotstop has not been found in Britain, and to complicate the matter further, if used in Britain, its host, Sitka spruce, is also non-native. Testing a non-autochthonous BCA on an introduced host raises a number of issues which are discussed below. A further complication when considering the use of *Phlebiopsis* for disease control is that the species *P. gigantea* is almost circum-polar in the northern hemisphere and is a single species, all isolates being capable of interbreeding. Furthermore, genetically distinct genotypes have developed in separated biota across the world [7]. Introducing a non-native genet into an existing, established population of different genets of the same species without understanding the consequences is no longer acceptable.

To be effective as a BCA in Sitka spruce stumps, Rotstop will need to compete as a saproxylic entity with the pathogen *H. annosum* for the woody substrate (xylem) that would otherwise be utilised by *H. annosum.* That is, it must decay xylem tissue (cellulose, xylans, lignin and a variety of resins and oleoresins) in freshly exposed stump surfaces. A problem would arise if, at the same time, Rotstop can colonise wounds on stems from which the anti-fungal agents in bark tissues were physically removed, as is often the case in thinning operations when skinning wounds on standing trees expose underlying xylem. The susceptibility of such wounds on Norway spruce in Finland to infection and decay from applied oidia of a local strain of *P. gigantea* was found in 1973 to be minimal [8]. No similar trials of the Rotstop isolate in Norway spruce have so far been reported [9], nor has any *P. gigantea* been tested on wounded Sitka spruce, albeit two possibly related species of *Peniophora* (namely, *P. dryina* (B. and C.) Rogs and Jacks., and *P. luna* Rom.), are recorded as causing brown rots on standing Sitka spruce in Queen Charlotte Islands [10].

Since Kallio’s (1973) trial, forest management systems have changed within Europe, the majority of felling and extraction now being performed by harvesting machines and wheeled forwarders. These, because of their size, must increase the likelihood for stems to be damaged. An added risk of infection by Rotstop occurs because the aqueous suspension in which Rotstop is held is applied to stumps by pressure systems mounted on felling machines [11,12]. These would inevitably release the BCA in aerosols that move through the forest on currents of air and are then positioned to invade fresh wounds on stems of damaged trees.

Both superficial and deep stem wounds are not uncommon in British plantations [13] of Sitka spruce and may result in local damage to xylem tissue, mostly from *Stereum sanguinolentum* Fr. Norway spruce is generally more susceptible to fungal colonisation of stem wounds than is Sitka spruce [14]. For example, after 12 years, rot had developed behind less than 1 per cent of deer-inflicted stem wounds in Sitka spruce in Scotland.

The work reported here forms part of a 1997–2000 assessment of the suitability of Rotstop for use on Sitka spruce stumps in Britain and was of a preliminary nature. Subsequent trials [15] reported by Tubby in 2010 reviewed its efficacy on Sitka spruce stumps in upland Britain, and the genetic variation within and between Scandinavian and U.K. isolates of *P. gigantea* (including the current isolates in PG Suspension) and Rotstop. These matters are discussed below.

Although this work was conducted over 20 years ago and was not reported until now, questions over the movements of BCAs have become a topic for discussion [7], and the peculiar situation within Britain as described above have made this a relevant issue.

## 2. Methods and Materials

In this paper, the term Rotstop describes the manufactured product that uses isolate No. VRA 1835 of *Phlebiopsis gigantea* (Fr.) Jülich collected from a spruce stump in Finland.

Trees were inoculated with oidial spore suspensions of Rotstop (to mimic aerial infections) or wooden dowels in holes drilled into sapwood to represent physical damage.

### 2.1. Site

Field work was conducted in a 44-year-old stand of pure Sitka spruce in Lille Hareskov Forest, (55°45′ N, 12°23′ E), Denmark. Average annual rainfall is 600 mm, and the soil is a silty-gley. When assessed in 1997, crop height averaged 22 m, with a mean diameter at 1.3 m of 27 cm. Forty co-dominant trees were selected from this crop in November 1997 and divided at random into four groups, each of ten trees. Trees were sampled after 11 to 18 months of incubation in the forest, each period containing at least one winter and part of a growing season.

### 2.2. Wounding

Seven treatments were applied to each tree (Table 1). Each tree received both treated and control inoculations, except for the P (Pressler boring), which was either treated or controlled. At each inoculation, each treatment was applied to all trees in a group before the next treatment was initiated. Treatments were applied in the order shown in Table 1.

The inoculation sites for spore suspension (S) and the wood dowel (W) treatments on the main stem were the 4 cardinal points around the circumference at 1.3 m above ground. Treatments were allocated to the stem inoculation sites at random. Two exposed lateral roots as far apart as possible were selected for inoculation with infected (R+) or uninfected (R−) 20 mm wood dowels. To avoid complications at sampling which could arise from overlapping stain columns, the Pressler boring inoculation site was selected to be approximately equidistant between the two root buttress inoculations, at a height of 0.5 m above ground level. The inoculation locations are shown in Figure 1.

Inoculation and sampling dates are shown in Table 2.

### 2.3. Preparation and Application of Inocula

Spores: Oidial spore suspensions were made on the days of inoculation by adding 5 g commercial Rotstop powder, ref VRA 1835, (minimum 5 × 10^6^ cfu) to 1 litre of sterile water, and shaking vigorously. Approximately 100 mls (0.5 × 10^6^ cfu) was decanted into a wide-mouthed, well-stoppered container and transported to the forest. The control consisted of sterile (autoclaved) water. A sterile 70 mm diameter filter paper was dipped into cold spore suspension until it was fully saturated (approx. 0.5 g of suspension), and adpressed to an 80 mm diameter decorticated wound surface (see below). Each filter paper carried approx. 2500 cfu of Rotstop (equivalent to 65 cfu per sq cm). A filter paper dipped in sterile water acted as a non-inoculated control. To provide shelter, a 100 mm × 100 mm square of heavy gauge polythene sheet was fixed over the wound with 10 mm staples driven into surrounding bark. The polythene was removed after 3 weeks.

Wood dowels: Two sizes of dowels were prepared: 20 mm diameter × 20 mm long dowels were punched longitudinally from the sapwood of transverse discs from a freshly felled Scots pine for use in W and R treatments, while 4.5 mm diameter by 150 mm long cores were extracted from the sapwood of fresh pine logs using a Pressler borer for the P treatments. Both types of wood dowels were sterilised and stored in a freezer until required. Approximately 2 months before each treatment, half were inoculated with oidial spores of the Rotstop isolate and incubated at room temperature, and the remainder were stored in a refrigerator as sterile controls.


Inoculation:


**P**. A single Pressler boring reaching at least as far as the pith and located 0.5 m above ground was removed from each tree using a 4.5 mm borer, following topical application of methylated spirits to the bark at the boring site. Trees showing evidence of butt rot were rejected, and alternative trees were selected. Immediately upon removal from the tree, the core was replaced by one which had previously been inoculated with Rotstop (P+, odd-numbered trees), or by a control core (P−, even-numbered trees). Cores were inserted to their maximum length or broken off level with the bark where they were longer than the cored hole. The entrance to the hole was sealed with petroleum jelly.

**W or R** (Wound or Root). Bark and phloem were carefully removed and discarded from an 80 mm diameter circle which was scribed into the bark around the inoculation point using a modified, sharpened tank-and-washer cutter mounted in a carpenter’s brace. At the locations of the W− and R− control treatments, a hole 20 mm deep was drilled with a 19 mm sterilised combination auger in the centre of the 80 mm decorticated wound. A sterile 20 mm wood dowel was knocked into this hole using a nylon hammer, and the exposed end was sealed with petroleum jelly. After all the W− and R− treatments were applied, similar wood dowels inoculated with Rotstop were inserted for the W+ and R+ treatments (see Figure 2).

After each inoculation, woody inoculants were tested for viability or sterility by plating out samples from each onto malt agar. The oidial spore suspension inoculum was checked by plating out drops onto PDA (potato dextrose agar). The viability of *P. gigantea* on treated filter papers was checked by re-isolation from unused test papers, which had been stored cold, 15 d after they were inoculated. These tests demonstrated that at each of the four inoculations, active Rotstop (*P**. gigantea)* had been applied to trees in wood dowels, borings or as oidial spores, and the analogous control treatments were sterile.

### 2.4. Sampling

All wounds were examined in detail after 11–17 months for signs of callous growth, and in the first sampling, for necrosis of phloem tissue. In each case, the wounds were prepared by carefully exposing surrounding phloem tissue with a drawknife. When the phloem had been assessed for the presence of necrotic tissue and had been photographed, the whole wound was shaved to expose the underlying xylem, from which four samples were removed axenically (using a sterile increment hammer) from around each inoculum dowel, at a regular distance of 25 mm from the nearest edge of the dowel, and a fifth from the dowel itself. The samples (20 mm long radially by 3 mm wide) were transferred directly into sterile self-sealing polybags from which they were later removed and were plated onto PDA. The conditions of the wounds were assessed, and each tree was felled, after which 25 mm thick transverse discs were cut from fixed positions above and below inoculation points on each tree. The transverse sections were washed and examined for the presence of stain associated with the wound and the inoculation. Axenic wood samples were removed from stained areas for culturing using an increment hammer. In culture, *P. gigantea* was identified by its gross morphology, the presence of oidia, the presence of clamp connections and by somatic compatibility tests.

Samples for TEM and SEM were obtained from sapwood blocks 30 × 20 × 7.5 mm, taken from around an infected dowel, and from 200 mm above and below the insertion point. From these, thick sections were cut, washed (pH 7.4), fixed, dehydrated in an ethanol series, mounted on aluminium slabs and sputter-coated in gold. For the SEM, samples were examined using a Cambridge Instruments S600 Scanning Electron Microscope operating at 7.5 v.

A further set of samples (0.5 × 1 × 1 mm) were embedded in resin, polymerised at 60 °C for 24 h, sectioned at 100–120 mm, stained and examined with a Philips 301 TEM at 80 kV.

We used a generalised linear mixed model (GLMM) to study the impact of different treatments on the length of stain, using the R package glmmTMB [16]. The response variable of this model is stain length with different treatments (P, W and R, but not S) as a fixed effect variable and date of inoculation and incubation period as random effect variables. The data were transformed by the square root method to enable comparison between different treatments. In addition, analysis of variance (ANOVA) was performed in R 4.2.1 [17]. The model was fitted to study the effects of different method of inoculation (mycelium and spores) on stain length and of different treatments on the success of re-isolation of *P. gigantea*. A post hoc Tukey honest significance test (HSD) was used to compare the methods and treatments.

## 3. Results

### 3.1. Numbers of Trees Sampled

Of the 40 trees originally selected, one from the fourth group died (from a combination of root killing by *Armillaria* and bark-beetle attack on the main stem) and is not included within the results, which were thereby derived from 39 trees (see Table 3).

### 3.2. Physical State of Wounds

The petroleum jelly used to coat the external surface of each dowel and the filter papers used for the inoculation of spores or water were, for the most part, intact and in their original positions when sampled 11–18 months post-treatment.

The exposed xylem in most wounds was covered with crusty, dry resin overlying a mixture of viscous resin and adhering dust. The 20 mm dowel inocula were still intact and in their original positions: those that had been inoculated with Rotstop were soft-textured and reddish-brown (Figure 3a). In contrast, the sterile dowels were firm, and their colour differed, being streaked dark brown to black (Figure 3b). The 4.5 mm Pressler borings were complete, though fragile.

The position of tongues of necrotic phloem above, below and to the side of each wound was recorded for wounds in the first assessment only, since the definition of necrosis was found to be imprecise. Necrotic phloem was characterised by a grey, water-soaked appearance with hard reddish brown to black edges. Typically, it extended axially 2–5 cm both up and down from affected wounds and was rarely more than 10 mm wide. Where necrosis extended inwards as far as the underlying xylem, the necrotic tissue overlays actively flowing resin deposits. There were no cases of tangential necrosis.

### 3.3. Re-Isolation of P. gigantea (Rotstop Isolate)

*Phlebiopsis gigantea* was reisolated from xylem in the vicinity of approx. 34% and 2% of wounds that had been inoculated with infected or sterile wood, respectively, and from 3% of spore-infected wounds (Table 4). The majority of re-isolations were made from distances of 2.5 cm or less from the site of the inoculation. The furthest distance *P. gigantea* was recovered 63 cm from its source.

Among inoculated wounds, *P. gigantea* was isolated significantly more often from columns of stain above and below wooden dowels/cores inserted into the main stem (W+ and P+) than above roots (R+) or wounds inoculated with spores (S+), *p* < 0.001. *P. gigantea* was absent from control wounds, with two exceptions. In each case, the column of stain associated with the wound resembled those from sterile wounds, and *P. gigantea* was not isolated other than at the inoculation point. This suggests an artifact of inoculation, possibly in the form of cross-contamination. *P. gigantea* was isolated axially from xylem above and below 2 of 78 (2.5%) control (sterile) dowels, but the appearance of the affected samples suggests a sampling error rather than a mistake in applying a particular treatment. There was no clear effect of inoculation time or length of incubation period on these characteristics when tested statistically. All isolations of *P. gigantea* made after felling were found to be somatically compatible with the original Rotstop isolate.

### 3.4. Extent of Stain Associated with Wounding and Inoculation

Columns of stain originating at an inoculation point were traced downwards on the surface of 25 mm thick discs cut at 10 cm regular intervals below inoculations and to a maximum height of 20 cm above the upper wounds. To aid assessment, stain was enhanced by spraying the disc surfaces with water. The mean measured axial extents of the columns for each inoculation and the number of columns measured are shown in Table 5. In each category, there were some columns of stain that extended more than 100 cm above the sampled limits; the results in Table 5 do not include these unmeasured lengths, and therefore represent an underestimate of the actual extents of stain within the sampled trees.

Results presented are more in the form of qualitative descriptions than accurately measured parameters, since determining the precise limits of vertical staining proved to be beyond the scope of the work. This is because the discolouration faded with increasing distance from the inoculation point and merged imperceptibly into healthy tissue, often between sampling points (see, e.g., Figure 3C).

Nevertheless, when data were analysed with GLMM in R to show the effect of treatments with mycelium inoculation (W+, R+ and P+) on stain length, significant differences could be seen (Figure 4). In particular, the W+ and P+ treatments caused significantly longer stains than other treatments, and although stain length for the R+ treatment showed a large variation, it was still significantly different from the R− (sterile wood dowel) treatment. All the wood dowel/bore core treatments (W+, R+ and P+) had significantly longer stain averages than the treatment with spores (S+) (*p* < 0.001 based on ANOVA and Tukey’s HSD test).

### 3.5. SEM and TEM Studies

Colonisation of these stain columns by the Rotstop isolate was confirmed by TEM studies. *P. gigantea* hyphae appeared to be able to spread rapidly between xylem tracheids through the penetration of pits and by formation of boreholes (Figure 3D). Large boreholes, with diameters many times greater than the hyphae within them, resulted from decay caused by hyphae passing through pits. In some cases, boreholes coalesced and thus increased the apparent rate of cell wall degradation. The formation of boreholes and subsequent T− or L− branching of hyphae within the S2 layer were probably responsible for the formation of cavities within the cell wall. Within tracheids, degradation of the S3 layer occurred initially, followed by the successive decay of the S2 and S1 layers and the compound middle lamellae, with the cell corners degraded last. This pattern suggests that *P. gigantea* was decaying cellulose and lignin simultaneously. This conclusion was further supported by the observation that cell wall degradation occurred in the immediate vicinity of hyphae [18,19] (see Figure 3E). The presence of erosion furrows [20] was also noted where fungal hyphae on the S3 layer had grown deeper into the cell wall.

### 3.6. Wood Moisture

Wood moisture, which was measured in fresh samples cut from trees from the first and last samplings, was expressed as percent saturation. Sapwood averaged around 70% of saturation, and heartwood, 28%.

## 4. Discussion

### 4.1. Implications for Wound Colonisation

With this trial, we sought to establish whether or not significant damage could be expected to occur to standing Sitka spruce if the non-native isolate of *Phlebiopsis gigantea* used in the BCA Rotstop were to be released in Britain. After approximately one year, *P. gigantea* was isolated from behind only one of 39 oidia-infused filter-paper inoculations. This indicated that this genotype, chosen for its ability to colonise Norway spruce stumps, can grow on intact, uninjured Sitka spruce sapwood that would be exposed in skinning injuries, but rarely. Survival of *P. gigantea* on these filter papers for at least 15 days post-treatment indicated that a viable inoculum had been presented to wounds. By contrast, infection of deeper wounds into sapwood was facilitated by drilling 20 mm radially into the stems and challenging the wounded sapwood to contact infection by Rotstop, which grew out of the 20 mm dowel inoculum into adjoining tissues in 33 of 97 (34%) inoculated wounds. These are broadly similar frequencies to those measured in the 1970s in Norway spruce [8]. Colonisation was further demonstrated by SEM and TEM studies on the growth of the fungus in small chips of infected sapwood removed from close to dowels [21]. These indicated a simultaneous lignin and cellulose degradation, starting with S3, followed by decay of S2, S1 and middle lamellae, and the cell corners degraded last. Some cell wall degradation occurred in the immediate vicinity of the hyphae. Such destruction of sapwood has been observed in spruce affected by the pathogen *H. annosum* [22] and may offer a clue as to the means by which *P. gigantea* excludes *H. annosum* from or overwhelms it in conifer stumps. There was no evidence that Rotstop infected any of the live callous tissue that developed after wounding.

The Rotstop isolate of *P. gigantea* also grew into heart- and sapwood above infected, inoculated 4.5 mm Pressler borings, indicating that it has the capacity to exist in moist (65–74% MC) sapwood and drier (28% MC) heartwood.

The staining of narrow zones of sapwood distally both above and below infected and sterile inoculum suggests that *P. gigantea* took advantage of the localised interruption of the vascular system of water transport that is inevitable if stems are wounded, in this case, if 19 mm diameter holes are drilled radially into living stems and if only periderm is removed. The likelihood is that axial drying-out was one contributory factor to the formation of stain columns which, in inoculated wounds, extended some distance up and down the stem. The development of analogous columns of stain among uninoculated control wounds, but to a considerably reduced extent, points to another, unknown cause, and in this case, we would suggest that severing water columns per se was contributory.

Wounding of live stems leads to many physiological or mycological changes, such as an increase in the elements P, Mg and K at the wound margins [23], while the role of stilbenes in resistance of fungal pathogens [24] show host responses that may be linked to the process of wound closure. Subsequent colonisation of such wounds in pine [25] showed that in 10-year-old tree transplants, Rotstop induced short-lived occluding necrosis within both xylem and phloem following wounding, in the process killing less cells than analogous *H. annosum* inoculant which, over the same time period, induced expanding necroses. Indeed, the protective effect of Rotstop appeared to go further by pre-challenging the tree’s response system into the formation of lignified bark necrosis. This suggests that in nature, *P. gigantea* might even confer some protection to open wounds. The issue of its role in standing trees remains open. Taken together, the damage caused from these inoculations does not suggest that Rotstop presents anything other than a minor risk to standing trees, and would certainly be very much less than that from the pathogen which the use of Rotstop should obviate.

However, this was a preliminary trial limited to one site and to 40 trees all the same age. It relied on within-tree variation to illustrate any effects of inoculation. In addition, it was limited to inoculation of fresh wounds with a single genotype of the BCA.

### 4.2. Wider Aspects of the Use of BCAs in British Forestry

Notwithstanding the limitations of this trial, it is sufficiently encouraging to consider other aspects of the potential use of this non-native BCA in Britain were it to be introduced for the control of *H. annosum* in British stands of Sitka spruce. Since the experiment was conducted in the late 1990s, significant advances have been made in understanding the evolutionary and genetic complexities of *P. gigantea* over parts of its range in Europe [26] and in North America [7], and its possible place within the diversity of monocultures of Sitka spruce in Britain and Ireland [27] over the long term. Together, these have raised concerns about the movement across evolutionary boundaries of *P. gigantea* destined for biocontrol. As noted in the Introduction, Sitka spruce is the single most important timber-producing species in Britain, whose contribution to the economy is valued in hundreds of millions GBP. Over time [3], significant butt rot in standing trees may impact its profitability, in which case its contribution to carbon storage, remission of climate change and species diversity may also be reduced.

The need for a prophylactic control became clear when the susceptibility of Sitka spruce to *H. annosum* was described in Britain [28]. Sufficient research into the epidemiology of the disease among first-rotation crops in upland Britain [2] permitted a decision-support system for guiding prophylactic control measures to be created [4]. Here, failure of urea [29] to provide adequate protection for future rotations of Sitka spruce and research into its mode of action [30] made the prospect of a biological analogue for stump treatment appealing. Testing a native British isolate of *P. gigantea* appeared to be the first obvious step to finding one for use on Sitka spruce in Britain. However, although effective on pine, early tests by Rishbeth (unreported) and the U.K. Forest Research consistently failed to control airborne spore infections by *H. annosum* on Sitka spruce stumps, even on dry, sandy sites in East England where there is a very high ambient spore load of *P. gigantea*. (Greig, Pers. Comm.) [31]. These trials, undertaken over some 40 years, involved other possible fungal competitors, including *Resinicium bicolor* (Alb. and Schwein.) Parmasto and *Melanotus proteus* (Kalchbr.) Singer [32]. All failed to provide acceptable control on Sitka spruce stumps [33]. In contrast, in Scandinavia (where spruce is native), evolution of *P. gigantea* from its primaeval host of pine into spruce stumps has provided a successful BCA for use on Norway spruce in Europe [6].

Nevertheless, in one trial on Sitka spruce stumps, Rotstop controlled 88% of infection by *H. annosum* in Davignac, southern France, in 1995 [34]. These excellent results stimulated further, generally unsuccessful trials of Rotstop in Britain [15]. The disparity of the French and British results deserve comment. The most obvious difference was likely to be stump moisture and temperature, both of which might affect the growth of Rotstop in Sitka spruce stumps [35]. Stump moisture was probably lower and temperature higher in France (no data on either available). In standing Sitka spruce, xylem moisture shows little variation over a year [36], whereas in stumps, moisture content near the stump surface was found to rise after felling [35] and continued to rise in succeeding years [37]. However, stump surfaces are a small part of the complete stump and its attached roots. Stumps are a notoriously variable source of inoculum [38] and in addition to their physiology are subject to mobilisation of resins and colonisation by fungi and bacteria, all of which can affect their susceptibility to infection by pathogens and saprobes.

Given these issues, the practicality of undertaking further stump treatment trials on Sitka spruce using Rotstop seems questionable. Before rejecting its possible use in Britain, however, it is worth considering that there are considerable economic advantages in working with a control agent that has international approval, since the costs associated with collecting the environmental and experimental data to support approval are prodigious. It follows that improving the application of Rotstop to stump surfaces is an obvious starting point [39], while using a breeding approach to improve efficacy is another [40]. A further potential area for research is the different susceptibility of Norway and Sitka spruce stumps to colonisation by both Rotstop and *H. annosum.* These two spruces diverged from a common ancestor 13.4 million years ago [41], and they have since become confined to opposite sides of the planet, and to very different ecosystems. Although classed as similarly susceptible to attack and decay by *H. annosum,* the type and pattern of butt rot within them is very different [28,42] and must represent a fundamental modification to the structure or components of the xylem of the two species. Interestingly, the increasingly widespread planting of Sitka spruce in Scandinavia is bringing these two spruces together again, with unknown consequences in the long term.

In terms of registration, development of a prophylactic BCA based on Rotstop for use in the U.K. is made more problematic by the fact that both the agent and the host would be non-autochthonous, the fungus deriving from a region (Scandinavia) where it co-evolved with a spruce that is non-native in Britain. That having been said, given continued concerns about the use of chemical-based pesticides, the deployment of biological analogues wherever possible seems prudent. For example, change in the distribution of invasive weed species, sparked by climate change, is stimulating the search for more effective and less risky BCAs for their suppression [43]. At the same time, it is becoming increasingly clear that a holistic approach should be taken towards assessing the long-term effects of BCAs before they are released into the wider environment. This is particularly the case when selecting BCAs that originate in a different geographic biota from the one in which they would be used. Under these circumstances, some form of quarantine for testing to ensure the long-term integrity of the receiving ecosystem is needed. Indeed, best practice might only allow BCAs that co-evolved with their potential hosts to be used [7]. Clarification is perhaps needed here on the objective of that constraint as it may apply to the widely distributed saprobe *P. gigantea*. If it is to maintain diversity within and purity of each of the members of the worldwide family of *P. gigantea* genets, then use of a BCA should be restrained to use within its natural genotypic and geographic range. If, on the other hand, the objective is to restrict its introduction to environments in which *P. gigantea* is already present, any mix of genotypes for use on co-evolved species might be acceptable. 

Given the constraint described above, it is relevant to consider the use of BCAs derived from fungal antagonists to *H. annosum* that have co-evolved with Sitka spruce in its native habitat. Limited observations suggest that *P. gigantea* has been detected amongst old-growth conifers within the native range of Sitka spruce in NW America, and if that is the case, the fungus and the spruce might then be classed as having co-evolved, in which case the use of a North American Rotstop isolate might be justified on Sitka spruce in Britain. The Pacific Northwest genets of *P. gigantea* might be quite different in some respects to those occurring naturally in Europe, even to the extent of describing them as a different species. More in-depth molecular work, perhaps whole genome sequencing, and comparisons are required before we mix these varied genotypes within the Northern Hemisphere. Indeed, it is our opinion that speciation within *P. gigantea* needs to be resolved before Rotstop or any other foreign genet is released into the British environment. This is because even experimental application of spores of a potential antagonist onto individual stumps can be seen as inoculative biocontrol, boosting deposition of naturally occurring competitors at the stump surface. In the long term, however, inoculative morphs into an inundative control as the BCA becomes established, and possibly dominant, within the wild biota. So, release of a novel biocontrol agent can be seen as a one-way, single event that cannot be reversed. This has implications for all subsequent tests of efficacy and related environmental impacts of Rotstop. There is one further twist to this issue. Current technology allows genotypic relatedness to be established for scattered geographical members of the species *P. gigantea* across the globe. Value would be added to those assessments if there was some way of assessing phenotypic relatedness as well through, e.g., gene expression, but with the obvious constraints described elsewhere [7]. Clearly, with a BCA as utilitarian as *P. gigantea* with its world-wide distribution and favourable mode of action against one of the most damaging tree diseases, further investment into uncovering more of its genetic make-up would be fully justified.

An additional problem peculiar to this form of BCA is to decide how such stump treatment efficacy trials can be conducted with adequate quarantine, especially within forests. Because of overriding edaphic and climatic influences of site on stump morphology and physiology, as referred to above, realistic tests of any biological stump treatment material can only be performed on freshly cut stump surfaces still attached to their roots in the forest [44], and not under controlled conditions such as in greenhouses. Where the potential BCA comes from the same biota as its host, undertaking such trials in forests presents fewer concerns. Herein lies a problem for finding a BCA for Sitka spruce in Britain if neither is native. At minimum, restricting application of the BCA to stump surfaces by droplet on still days and fully harvesting all colonised stump material before the fungus can fruit (on small pine stumps within a year) would seem prudent. Large-scale trials involving harvester head spray application [11] in which spores of Rotstop would be uncontrollably disseminated in high-pressure sprays should, naturally, be avoided.

Another area of concern is the impact that release of Rotstop into the alien environment of the U.K. might have on local populations of *P. gigantea* and, in the long term, on the developing ecosystems that are British and Irish forests. Here, it is necessary to recognise the essential difference between new forests in the U.K. and those long-established or natural stands in Europe and in NW America, the latter including old-growth stands which have never been intensively managed and which are the ancestral home of Sitka spruce. The majority of U.K. stands of Sitka spruce are less than about 60 years old, first established in the 20th century on cool, wetter uplands where no trees had been growing for hundreds of years. Soils here are typically nutritionally depleted, acidic (pH < 5), poorly drained and (one would imagine) deficient in diversity of, inter alia, soil and arboreal fungi. Of the latter, *P. gigantea* has been notable for its absence from the thousands of Sitka spruce stumps sampled for studies into *H. annosum* throughout Scotland in the 1980s and 1990s by U.K. Forest Research and others, regardless of the fact that large numbers of oidia have been applied every working day for 50 years in the pine forests of East England (400 km) and Northeast Scotland (200 km) to the east. However, recent studies [45,46,47] suggest that such depletions in wildlife that were evident when these stands were first established are changing much more rapidly than thought possible. Allied to that, new ways of managing them [48] along with expected climate change will, in the long term, have significant effects on both their diversity and (perhaps surprisingly) on their susceptibility to infection and colonisation by *H. annosum*. Set against this uncertain backdrop, it is instructive to consider what might happen when these relatively young plantations mature as stable ecosystems. For example, it is clear from work conducted in the forests of Northern Europe [49] that Rotstop applied to Norway spruce stumps has been recovered from those stumps up to six years later, whereas in pine stumps, replacement of the Rotstop isolate by wild isolates of the same species, or by other species, seemed to prevent the formation of monomorphic populations of Rotstop in treated stands [26].

Introducing an alien fungus into a newly emerging forest ecosystem requires tests to ensure both effectiveness and environmental safety. Both need to be performed in the field, as explained above, but neither should, if the protocol banning the use of BCAs that did not coevolve with their hosts is employed. How and when that hurdle may be solved is beyond the scope of this paper.

## 5. Conclusions

It seems unlikely that the current formulation of Rotstop would pose a problem for the health of British stands of Sitka spruce. However, we believe that a long-term holistic view of the place of Rotstop among emerging and changing habitats within British forests must affect any decisions to import Rotstop into the U.K.

## Figures and Tables

**Figure 1 pathogens-11-00937-f001:**
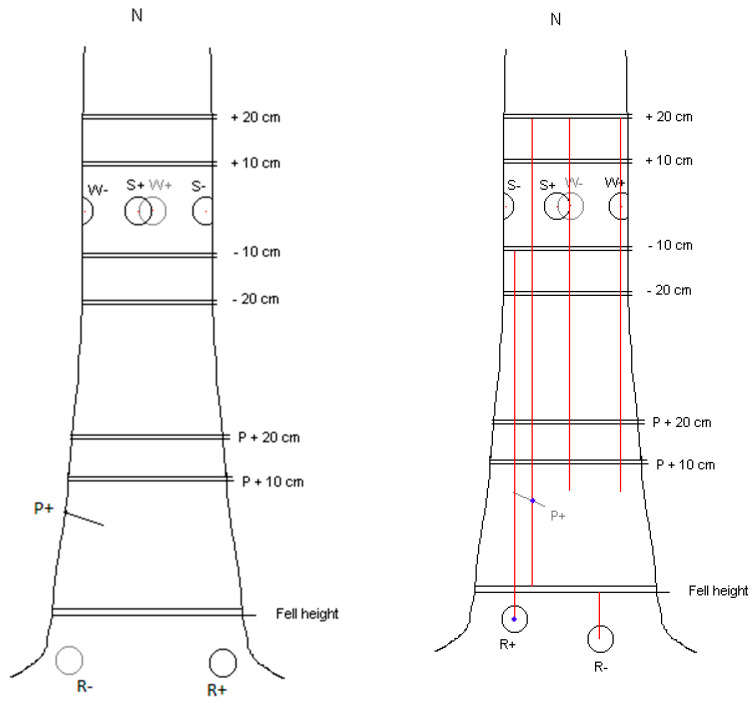
Principles of placement of the inoculation points on a tree. Also shown are the sample points where discs were taken above and below stem wounds (S and W) and above the bore core (P) and root (R) inoculations. To the right is how stains were tracked from actual inoculations on each tree via discrete stain columns observed on the sample discs, in this case tree no. 23. No stains were traced further than the W/S + 20 cm discs. Note the extent of the stain columns upwards from the R+ and P+ inoculations.

**Figure 2 pathogens-11-00937-f002:**
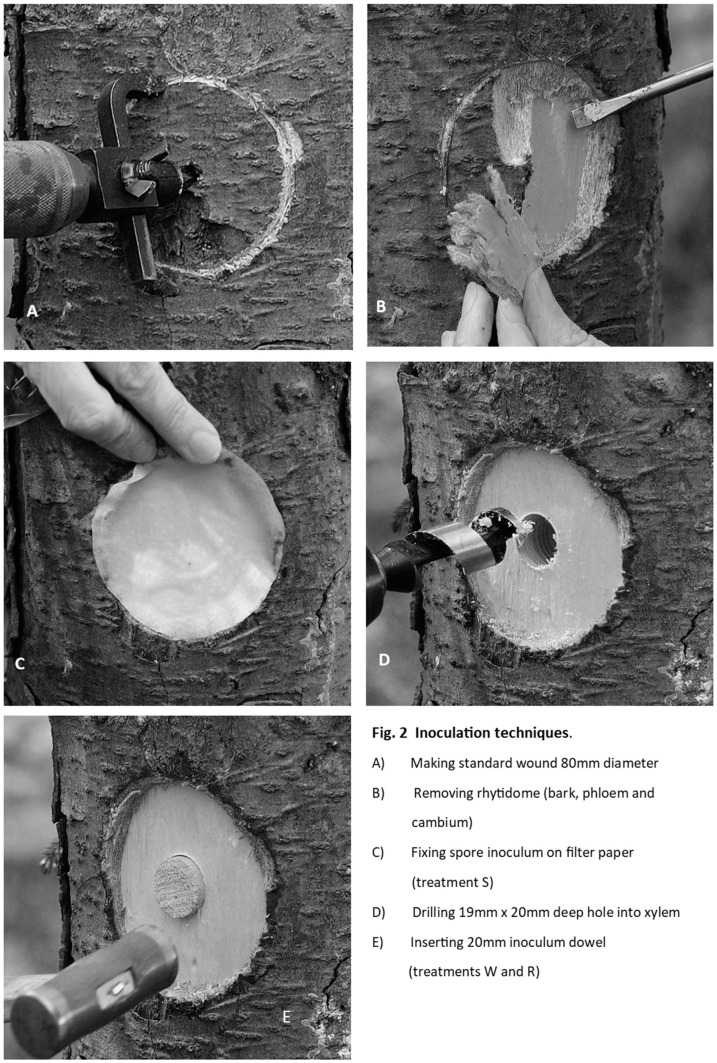
Method used for preparing inoculation sites by decortication, drilling and insertion of inoculum dowels or filter-paper.

**Figure 3 pathogens-11-00937-f003:**
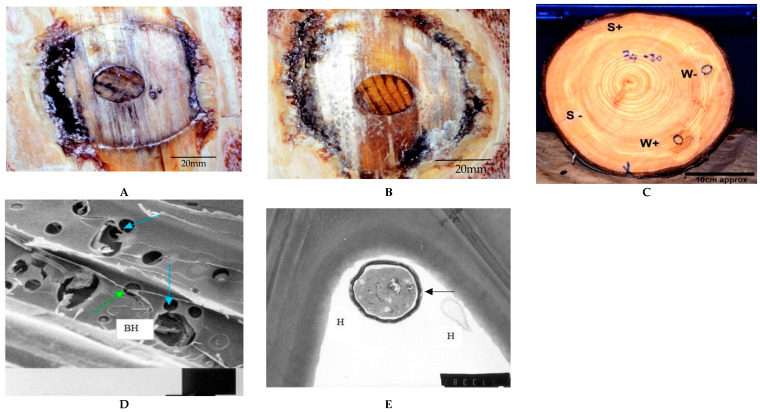
Tree 34, 17 months post-inoculation. (**A**) Sterile W– inoculant. P. gigantea was not isolated from this wound, which produced limited staining. Grey-black discolouration of the dowel typical of W– and R– wounds. (**B**) Rotstop-inoculated W+. P. gigantea (Rotstop variant) re-isolated from xylem above and below this wound. (**C**): Cross-section disc 20 cm below W and S inoculations. Note lack of staining below S−, S+ and W– inoculations but present below W+. Note also annular straw-coloured dry zone which normally defines heartwood deviating around both dowel inoculations. (**D**,**E**) SEM and TEM of samples of wood taken from above and below W+ (reproduced with permission of Peter Bailey). (**D**) Penetration of pits (blue arrows) by hyphae (green arrows) has resulted in the formation of large boreholes (BH) within the cell wall (bar = 20 µm). (**E**) Hyphae (H) of P. gigantea growing in the lumen on the S3 layer have caused localised degradation of the S3 layer (black arrow), possibly due to enzymes released by the mucilage coating around the hyphae (bar = 1.25 µm).

**Figure 4 pathogens-11-00937-f004:**
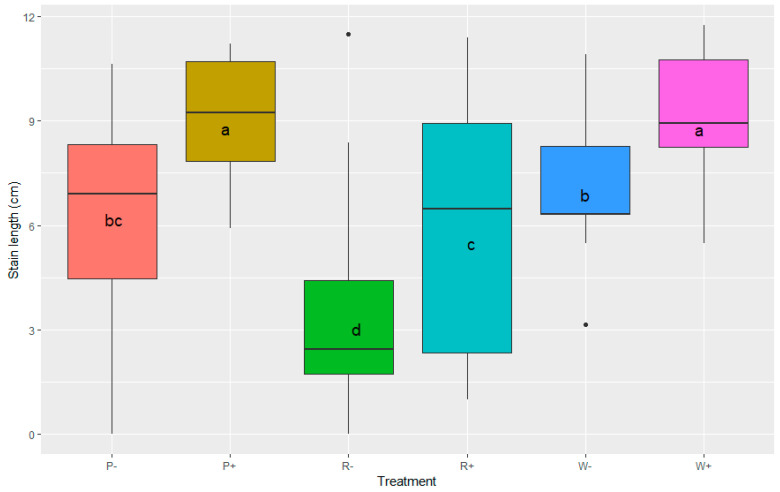
Box plot representing the effects of different inoculation treatments (P, R and W) on the stain length measured in cm. A Tukey post hoc test revealed significant differences of LSmean among treatments, represented by letters. The difference between a and b is significant at the *p* < 0.01 level, and the difference between d and all other letters (a, b, c) is significant at the *p* < 0.001 level. The centre bars of the box plots represent the median.

**Table 1 pathogens-11-00937-t001:** Treatments.

P+ or P−	S+ and S−	W+ and W−	R+ and R−
4.5 mm Pressler boring colonised by Rotstop (P+) or sterile (P−) inoculated into radial hole drilled to tree centres.	Oidial spore suspension (S+) or sterile water (S−) on filter paper placed onto exposed xylem in the centre of an 80 mm circular wound.	20 mm × 20 mm wood dowel colonised by Rotstop (W+) or sterile (W−) inserted into 19 mm diameter radial hole at the centre of 80 mm diameter stem wounds.	20 mm × 20 mm wood dowel colonised by Rotstop (R+) or sterile (R−) inserted into 19 mm diameter radial hole at the centre of an 80 mm diameter wound cut into the surface of a lateral root.

**Table 2 pathogens-11-00937-t002:** Felling and sampling dates.

Inoculation Number	1	2	3	4
Inoculation date	20.11.97	13.4.98	7.7.98	29.9.98
Felling and sampling date	21.10.98	20.5.99	20.5.99	6.3.2000
Months incubated	11	13	10	17

**Table 3 pathogens-11-00937-t003:** Sampled wounds.

Inoculum Type	Treatment	Inoculum Position	+Rotstop	−Rotstop	Number Inoculated
Oidial spores	S+, S−	Stem	39	39	40
20 mm pine dowel	W+, W−	Stem	39	39	40
20 mm pine dowel	R+, R−	Root	39	39	40
4.5 mm pine core	P+, P−	Stem	19	20	20 *

* Note: Only one P treatment per tree.

**Table 4 pathogens-11-00937-t004:** Frequency of re-isolation of *P. gigantea* (Rotstop isolate) from inoculated and uninoculated wounds, and, in brackets, maximum extent of colonisation above (+) or below (−) the wound which was determined by re-isolation. Different letters indicate significant differences between treatments in re-isolation success.

Inoculation	W+	W−	R+	R−	P+	P−	S+	S−
Total	15/39	2/39	7/39	0/39	11/19	0/20	1/39	0/39
Percent	39% ^a^	5% ^b^	18% ^b^	0% ^b^	58% ^a^	0% ^b^	3% ^b^	0% ^b^
Max extent, cm	(+20/−51)	(0)	(+63/−0)	(0)	(+10/−27)	(0)	(0)	(0)

**Table 5 pathogens-11-00937-t005:** Mean measured extent (cm) of stain columns originating from wounds.

Treatment	S+	S−	W+	W−	R+	R−	P+	P−
Mean axial extent (cm)	8	14	85	52	45	18	80	52
No. columns of stain	9	14	39	38	39	36	19	19

## Data Availability

Not applicable.

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
