# Peer review of "An Experiment in Denmark to Infect Wounded Sitka Spruce with the Rotstop Isolate of Phlebiopsis gigantea, and Its Implications for the Control of Heterobasidion annosum in Britain"

_pathogens, 2022, doi:10.3390/pathogens11080937_

Round 1
Reviewer 1 Report
This manuscript provides interesting data on the colonisation of Sitka spruce stems following inoculation with the RotStop strain of Phlebiopsis gigantea, as early tests for any possible complications in applying this saprotrophic fungus as a biological control agent in UK forests.
I have very few comments on the content of the manuscript, although there are some formatting points that (may) require addressing. Is it journal policy to use ‘ibid’ in referencing? I thought that system had been replaced when we went over to referencing in lists rather than as footnotes! Sadly, I doubt many current readers would understand what it means.
The methodological approach is sound and the field inoculations are described well. There is a useful section in the results however, which describes the patterns of decay observed using transmission- and scanning-electron microscopy, but there are no methods given for this part of the work (they are probably given in the cited paper, but as currently referenced, that would be difficult to find.
The complex considerations that need to be addressed are discussed very nicely and the conclusions are sound. For example, Pacific Northwest genets of P. gigantea might be quite different in some respects, to the P. gigantea occurring in naturally in Scandinavia (or the UK), even if not sufficiently so to describe it as a separate species. I suspect that more in-depth molecular work, perhaps even whole genome sequencing and comparisons, are required before we spread these different genotypes widely in the Northern Hemisphere. This point is adequately discussed in the remaining part of this paragraph – very interesting.
Rotstop vs P. gigantea in Denmark
The specific name of Phlebiopsis gigantea in the tile should begin with a lower case letter.
I suspect the author affiliations are presented in the wrong order!
Lines 14-15: Is it the Rotstop ‘strain’ of P. gigantea that is native to Britain? Different isolates of the same potential BCA undoubtedly have varying effects. [you claim it is not native to Britain in the introduction].
Line 27: the possessive (‘) is not required on BCAs – it is plural.
Line 29: Is it journal policy to use authorities in the abstract?
Line 58: ‘L.’ should not be in italics.
Line 59: ‘saprotrophic’.
Line 65: ‘long-term’ used twice?
Line 93: Although harvesting machines and forwarders might increase the ‘opportunity’ for stem damage, I think ‘possibility’ is a better word.
Table 1: make sure there’s a space between numbers and units. It is variable in this table.
Line 132: How can root inoculations be made at 0.5 m? I understand that, for a temperate tree, Sitka spruce produces quite large buttressing roots. Would it be more appropriate to state how far from the root collar/flare the inoculation points were?
Line 144: get rid of the apostrophe after CFU! (Same problem on lines 149 and 150). There appears to be an errant full stop at the end of line 144.
Table 2 – clearly the work was carried out some time ago! That in no way affects the validity of the results.
Line 201: ‘potato dextrose agar’ was defined as PDA (or the other way around, in fact) on the previous page.
Line 238: ’63 cm from its source’. Do you mean from the inoculation point?
Lines 255-260: In inoculations with Heterobasidion, a quantitative PCR-based method showed that the pathogen had grown well beyond nay signs of necrosis in the (limited) sapwood present (Bodles et al. 2006) or the ability to reisolate the fungus from plated stem discs. I wonder if the reality with P. gigantea is similar?
Table 5: no statistical analyses on these data?
Lines 270 – 284: Nicely presented, but I do not remember seeing any methods for the SEM or TEM.
Figure 3: The text font changes for D and E.
Line 304: The Kallio reference should be cited as a number in brackets.
Lines 323-325: as long as no sapstain fungi managed to get into the wound.
Line 385: ‘…colonisation by fungi and bacteria etc…’ – what is ‘etc’? If there are other possibilities, they should be listed, as far as is possible.
Line 410: ‘…less risky BCA’s…’ – surely it is a plural, hence BCAs? [same on lines 412, 416, 485]. There are other examples of this issue.
BODLES, W.J.A., FOSSDAL, C.G. & WOODWARD, S. (2006). Multiplex Real-Time PCR detection of pathogen colonisation in the bark and wood of Picea sitchensis clones with different levels of resistance to Heterobasidion annosum. Tree Physiology 26: 775-782.
Author Response
Thank you for your helpful comments, all of which (with one exception) have been followed. The exception is the inclusion of Bodles paper. The work was designed to provide a fairly gross estimate of the effect of inoculation, and we did not have the resources to extend the sampling beyond that which was undertaken. Although potentially interesting to learn about the colonisation of these wounds in fine detail, the important observation we made was that Rotstop can decay Sitka spruce , which had not been shown before.
Reviewer 2 Report
Interesting paper with data which looks properly analyzed. Few notes are on the attached file

Author Response
Thank you for your comments, all of which were used
Reviewer 3 Report
Dear author please check if affiliation with e-mail address is well appointed to the authors
Abstract
For abstract read the instruction of journal: The abstract should be a single paragraph, a total of about 200 words maximum, need to be rewritten
l. 15-16: check paper Astra Zaluma et al. 2019
Introduction
The content and the extent of this chapter is appropriate, well written, contains basic information related to the topic, but I suggest to include some short paragraph about already known results related to application and use of P. gigantea for spruce treatment, especially Sitka spruce
L. 34-40 need refences
l. 49- three, not tree stump?
l. 51-53 reference needed
l. 53- if the Latin name occurred first time in the main text write out the genus name
l. 60 Rotstop- I suggest, for example in brackets, shortly include information about Rotstop, similarly as in the Abstract; I found more suitable to explain the term here, in the main text, or both also in
l. 62-64: include reference, currently not clear to which country the information belongs - Britain or Scandinavia
Abstract
l. 62, 74 –a space is missing before bracket -reference
l. 68 BCA- is the abbreviation explained in the text above?
last paragraph l. 104-107it is surprising that results are going to be published after about 20 yrs. The scientific aim / aims need to be clearly defined. Currently it is not clear why they are going to be published now, and is not evident what is the originality of the present work (comparing with Tubby 2010), if the results were reported in 2010
are these data, results still topical and original? Support originally and up-to-dateness of your results,
some results have been already published for example Astra Zaluma et al. 2019 in For Pathology
furthermore, discussion with other relevant published paper is set chapter for research articles, to discuss your results with previously published is defined part, chapter of each scientific article, it is not real scientific aim
l. 107 italics P. gigantea
clarify – why the experiments were realised in Denmark, not in Britain, for experimental use it is possible to ask permission to use BCA
Methods
Name of this chapter – include also word “Materials”, what is also described here
l. 118- Table 2 is mentioned earlier in the text as Tab 1, they have to appear in consecutive order
l. 123 to be clear what kind of 7 treatments, add information that P+, P- were applied only one per tree, no both treatments as in case of the rest of treatments
fig 1- include P-, for example like possibility: P+/P-
l. 160- include conditions for incubation
l. 162-163: already said in lines 122-123, this is repetition,
l. 165- the size of borer is 4.5mm or in table 4.3mm, explain
l. 182-183 what is the reason that you chose different type of media, for woody inoculant and spore suspension
l. 183- result? Not methodological part
l. 188 – please rewrite, referring to Tab
Tab. 2 has no legend
l. 201- use abbreviation PDA
there is a chapter SEM and TEM studies, Wood moisture in results - a corresponding chapter is missing in Materials and Methods
Results
l. 213 and Tab 3- 39 trees were evaluated, if only 1 tree died, for treatment P+/P- one tree is missing in Tab in one case of positive or control treatment
l. 234: are the calculation correct? 33%, it is for W+: 15 from 39?
l. 251-252: how you examined this effect, not mentioned in M&M, which statistical method?
l.286-287- these are methodological aspects, move to M & M chapter, keep here only results
Discussion
I found the Discussion too long and should be reduced. Only the issues that were studied should be discussed, needs to be reedited. Please include recent, newly published papers from last 10 years related to topic. Furthermore, there is a lot of text (speculations, assumptions) not supported be reference, please include reference or rewrite or delete those parts, for example there is no reference in the paragraph starting in l. 424 or lines 457-470
l. 299: at least 15 days? Not mentioned in M&M, this is results of reisolation; one reisolation was after sampling 10-17 months after inoculation, but 15 days? If it is not results of your current study include reference
l. 396: common ancestor 13.4mya? not clear what is 13.4mya
l. 416: number the reference Evans 2013
l. 461: natural stands in EU, it is imported, planted species in EU
References
Please carefully check the refence list to avoid duplicity, for example Redfern et al 2010, numbers 2 and 29 in reference list, or numbers 15 and 39, Pratt 2005: 3 and 27 in the refence list
Check the correctness of author´s name: for example, “Tubby” and “Tuby”
Dear Editor and Authors
I doubt about the meaning and relevance of publishing this article, since the knowledge is far onwards, experiments with Sitka spruce stump inoculation already realised. I suggest authors to clearly state the originality, novelty and importance of this experiment for the research topic, otherwise I suggest to rewrite the article to review type. The Discussion would be a nice basis for the review article, where you could include also this your unpublished result. The clause in Abstract (l. 24-27) and the topic developed in Discussion also imply to review type of article.
Author Response
Thank you for your comments, all of which were reasonable and thought-provoking. Most have been taken on board. Two issues remain. I have not reduced the abstract because I want it to be informative in a journal in which one of the aims is to stimulate debate. the other three reviewers were satisfied with it. Second , and for the same reason, I have not revamped the entire article to make it into a review. The experimental work described was unique and, from the British point of view, crucial to any decisions that we make about the control of H.annosum in the long term, and it provides context for the discussion in which these issues are discussed for the first time. This particular volume is tightly-targetted at tree diseases and their prevention, and I believe (as do the other three reviewers) that it does that.
But I am grateful for your views, which I understand and respect although on this occasion I disagree with them.
Reviewer 4 Report
Manuscript 1802846 submitted to Pathogens by Iben Thomsen & Pratt evaluates the possibility that the biocontrol agent Phlebiopsis gigantea may cause decay in wounded Sitka spruce. No data exist for this host species and no so many on Norway spruce. The work is well conducted with an adequate number of replicate spread over 4 inoculation dates. The manuscript is clear and well written and method are properly described. The discussion on the risk that may represent an alien isolate of a native species is quite interesting. I do not have much remarks to do. Some are made directly on the pdf.
The main one is that no statistical analysis is present for the frequency of P. gigantea isolation. This should be improved. A logistic regression with main effect of inoculation treatment, P. gigantea / water and inoculation date with the tree in random factor should be adequate. A number of software can be used (glmmTMB package from R for example).
Minor comment:
L71:72. Not sure I understand. You may what to clarify
1. the accronym BCA is not explained. Is that biological control agent ?
2. P. gigantea is not native to UK? Or is it just that it is an alien genotype of a native species
L77: I had understood L51 that PG was not efficient as a biological control on sitka spruce. I this not the case? From the discussion, it appear to be unclear. Maybe that the report are conflicting here would clarify.
L326-328. Change in water saturation / oxygenation that enables already present spores of wood decay fungi to germinate and start to decay?
Author Response
Thank you for your very helpful comments, as a result of which we have undertaken statistical tests and reported them. the paper is much improved as a consequence.